# Assessing motivators for utilizing family planning services among youth students in higher learning institutions in Dodoma, Tanzania: Protocol for analytical cross sectional study

**Upendo Munuo** [1] *, **Fabiola Vincent Moshi** [2]

**1** Department of Clinical Nursing, School of Nursing and Public Health, The University of Dodoma, Dodoma, Tanzania, **2** Department of Nursing Management and Education, School of Nursing and Public Health, The University of Dodoma, Dodoma, Tanzania

* pendomunuo@yahoo.com

## Abstract

### Introduction

Contraceptive services utilization is an important intervention in averting the impact of unwanted and unplanned pregnancy among youth which is an obstacle to the higher learning institutions youth students in attaining their educational goals. Therefore, the current protocol aims to assess the motivators for family planning service utilization among youth student in higher learning institutions in Dodoma Tanzania.

### Methods

This study will be a cross-sectional study with quantitative approach. A multistage sampling technique will be employed in studying 421 youth students aged between 18 to 24 years using structured self-administered questionnaire adopted from the previous studies. The study outcome will be family planning service utilization and independent variables will be family planning service utilization environment, knowledge factors, and perception factors. Other factors such as socio-demographic characteristics will be assessed if they are confounding factors. A factor will be considered as a confounder if it associates with both the dependent and the independent variables. Multivariable Binary logistic regression will be employed in determining the motivators for family planning utilization. The results will be presented using percentages, frequencies, and Odds Ratios and the association will be considered statistically significant at p-value $<0.05$.

**Data Availability Statement:** Deidentified research data will be made publicly available when the study is completed and published.

## Introduction

World Health Organization (WHO) defines youth as the phase in which an individual is no longer a child but not yet an adult. Age wise, adolescence is regarded from 10 to 24 years [1].

**Funding:** The authors received no specific funding for this work.

**Competing interests:** The authors have declared that no competing interest exist.

Globally, 1.2 billion out of the estimated 3 billion young population are aged less than 25 years [2]. In sub-Sahara Africa, nearly 1 in every 3 individuals is aged between 10 to 24 years [3]. Young girls are the most vulnerable group as they do not use sexual reproductive services including family planning, yet they are given little attention, especially in sub-Saharan Africa. This is due to the fact that they are the victims of early marriage, teenage pregnancy, unwanted and unplanned pregnancy, and sexual transmitted infections associated with early debut to unsafe sexual behaviors [4].

In low and middle income countries like Tanzania one-third of the women are wedded at the age below 18 years, while one in nine are married at 15 years old [5]. It is estimated that 11% of the births occur among adolescents worldwide whereby more than 90% of these births occur in low and middle income countries [6]. Sub-Saharan Africa is among the regions with highest teenage pregnancy rates but with the lowest rates of family planning (FP) uptakes [7].

In Tanzania, particularly, adolescents comprise almost one quarter (24%, or 12.8 million) of the total population [8]. The issue of health services provision such as reproductive health, nutrition services and screening services to young people is a concern, which requires attention of the government and different stakeholders. Increasing utilization of sexual and reproductive health services (SRHs) including FP services among adolescents is a crucial strategy in averting teenage pregnancies, unwanted and unplanned pregnancy, unprotected sexual intercourse which can result to reduction of maternal morbidity and mortality and STIs risks [7].

Promoting FP services use among adolescents is vital in achieving universal access to sexual and reproductive health services [9]. Sub-Sahara Africa (SSA) has the greatest unmet need for family planning use [10], and young people were reported to under-utilize FP services compared to other groups of women of the reproductive age in the region [7]. In connection to that, International Center for Research on Women (ICRW) analysis report that the unmet need for family planning use is greater among unmarried adolescents than those who are married, though married adolescents aged 15–19 experience a higher proportion of unmet needs compared to all married women [5].

Regarding the report of International Conference for Population and Development (ICPD), user-friendly sexual and reproductive health (SRH) services to adolescents is a paramount agenda [11]. User-friendly SRH services offers adolescents an opportunity to get accurate reproductive health information, education, counseling and health promotion activities [12]. In addition, adolescents regard SRH services to be user-friendly if the particular service maintains respect and confidentiality [1]. According to WHO guideline entitled *"Making Health Services Adolescent Friendly"* SRH/FP services targeted for them should bear the quality elements which include: accessibility, acceptability, equitability, appropriateness, and effectiveness [1]. Adolescents need to be offered FP services in the environment which favors their youth rights and should feel welcomed by health care providers [13].

In Tanzania, particularly in Dodoma, the available and recommended FP methods for youth and young girls are oral contraceptives pills such as Combined oral contraceptives (Microgynon), Progestin-only pills (Microval), Emergency contraceptives, Injectables such as Depo medroxyprogesterone acetate (DMPA), Implants that include Double-rod implant (Jadelle) and Single-rod implant (Implanon), Intra uterine devices (Copper T 380A) [14]. Additionally, there are natural methods such as Standard Days Methods (SDM), Lactational Amenorrhea Method (LAM) and Cycle Beads. Also, barrier methods such as male condoms and female condoms offer both protection against unwanted pregnancy and sexually transmitted infection including HIV/AIDS (dual protection) [14]. However, those family planning methods if consistently and correctly used, will prevent unplanned or unintended pregnancies and thus curtailing preventable maternal morbidity and mortality amongst adolescent girls [14]. It is predicted that universal access and utilization of family planning methods amongst

adolescents could lead to a decrease of 2.1 million unplanned and or unintended births, 3.2 million unsafe abortions, and 5600 maternal deaths each year [15, 16]. Utilization of family planning methods further provides adolescents with an opportunity to make informed decisions about when they want to have their children [17].

The national target for FP services utilization among women of childbearing age (15–49) including the youth, is 60% [18]. The nation also aims to reduce teenage pregnancy up to 20% by 2025 [19]. The available evidence indicates that only 18.9% of youths aged between 20 and 24 years use FP services against the national average of 32% (TDHS, 2004/05; 2010; 2015/16). The teenage pregnancy rate is also high, 44% [20].

With respect to studies conducted elsewhere among youths in higher learning institutions in Tanzania, the coverage is still low, irrespective of all these initiatives. For example, it was found that only 47.4%, 33.2% and 16% of the youths were using FP services at St John's University of Tanzania, universities from Dar es Salaam Region, and universities from Kilimanjaro Region, respectively [21–23]. Evidence from studies conducted outside Tanzania also reported a similar trend: Uganda (46.6%) and Ethiopia (16.2%) [24, 25]. In addition, evidence from other studies among youth from universities reveals a high prevalence of sexually transmitted infections, including HIV/AID [26–28].

The youths from higher learning institutions tend to engage in unplanned and unprotected sexual activities easily, which is thought to be attributed to by lack of strict parental environment [23]. This behavior may subject the youth to unintended pregnancies, abortions and sexually transmitted infections [21–23, 29]. Similarly, unintended pregnancies may bring in social consequences to the students, such as school dropout, economic hardships, social abandonment and neglect [30]. Moreover, evidence from the study conducted in universities from Dar Es Salaam Region revealed that unwanted pregnancy rate was 27% and abortion rate was 54.6% [23]. However, it is not well known what might be the motivators for university youth students to use FP services in Tanzania, particularly in Dodoma region. Similarly, majority of the studies focused on female undergraduates, while others confined themselves to a single institution [21, 29] Therefore, the broad objective of this study is to assess the motivators for family planning services utilization among higher learning institutions youth students in Dodoma Tanzania. To achieve this broad objective, the specific objectives are:

1. To determine the prevalence of FP service utilization among higher learning youth students in Dodoma Tanzania

2. To determine the level of knowledge of FP services among higher learning youth students in Dodoma Tanzania

3. To determine the perception of FP services utilization among higher learning youth students in Dodoma Tanzania

4. To assess the FP utilization environment among higher learning youth students in Dodoma Tanzania

5. To assess the motivators for FP services utilization among higher learning youth students in Dodoma Tanzania.

The findings of this study are expected, to bridge knowledge gap by addressing motivators for family planning utilization among youth students in higher learning institutions in Dodoma Tanzania. This will help to improve the utilization of family planning services. Moreover, the findings will help to develop different strategies, which will increase utilization of family planning services among youth students from higher learning institutions in Dodoma Tanzania. Furthermore, the findings will act as catalyst for other researchers to conduct more

in-depth research on the utilization of family planning services among youth students in higher learning institutions in Dodoma Tanzania and elsewhere in the world.

## Methods

### Study area

The study will be conducted in Dodoma City. Dodoma city has the size of 2769 square kilometers. It has: 453,844 households, 41 wards 170 streets 42 villages and 89 hamlets. The Dodoma city is located at the south eastern end of the Tanzania Central Plateau at an elevation of 1,200 meters above sea level. The town is located at the geographical center of the country on the vital Central Railway line; and on major crossroad of the National East West trunk road and the famous north to south Cape to Gairo Great North Road, which passes in Tanzania through Mbeya, Iringa, Dodoma, Babati and Arusha. The town is 465 kilometers from Dar-es-Salaam, and 425 kilometers from Arusha situated at 6oo south of the Equator. It has a total population of more than 2.6 million with growth rate of 3.1% [31, 32]. The total fertility rate in Dodoma city is 5.7% [20]. There are about eight (8) higher learning institutions in Dodoma city, which include: the University of Dodoma (UDOM), St John's University of Dodoma (SJUT), College of Business Education (CBE), and Institute of Rural Development Planning (IRDP). Others are Local Government Training Institution Hombolo—Dodoma, Capital Teachers College, Dodoma Institute of Health and Allied Sciences (DIHAS), and DECCA College of Health and Allied Sciences. Majority of these learning institutions provide RMNCAH services, including free family planning services. Higher leaning institutions in Dodoma city were selected because studies document that students from these institutions in sub-Saharan Africa are reported to indulge in unplanned and unprotected sexual intercourse, which risk them to teenage pregnancies, unsafe abortions and STIs [33, 34].

### Study design

This will be analytical cross sectional study design underpinned with quantitative approach to assess the motivators for family planning services utilization among higher learning institution youth students in Dodoma region, Tanzania.

### Study population

The study population will be all youth students from the selected higher learning institutions in Dodoma city and youth students aged between 18–24 years in the first to final year from the selected institutions will be included in the study. All youth students (aged 18–24 years) who will be married, sick or well but unwilling to participate in the study will be excluded from this study.

### Sample size calculation and sampling technique

The Cochran's, (1977) [35] formula will be used to determine the minimum sample size required for this study.

$$n = \frac{z^2 P(100 - P)}{d^2} = \frac{1.96^2 \times 47(100 - 47)}{5^2} = 383$$

Where:

■ n = desired sample size.

■ z = critical value at 95% confidence level corresponds to 1.96

- d = marginal error (desired level of precision which is 5%).

- P = prevalence of FP use among youth students (47%) [21].

To cater for potential attrition (unresponsiveness) 10% of the minimum sample size will be added up as shown below:

- $10/100^*383 = 38.3 \approx 38 + 383 = 421$

Therefore, the total sample size for this study will be 421 participants.

To avoid over representation of the population and reduce the cost of sampling in terms of time, manpower and monetary cost, multi-stage sampling will be employed to select participants from higher learning institutions in which three stages will be involved.

- **First stage** four out of eight higher learning institutions available in Dodoma city will be selected by simple random sampling technique using lottery approach.

- **Second stage** After the selection of the institution, if the institution will have more than one college or faculty, such as in UDOM and SJUT one college or faculty will be selected randomly using a lottery approach.

- **Third stage** After the selection of four institutions, a proportionate stratified sampling technique will be employed to determine the sample size from each selected institution to give a total of 421 participants. This will work out in the following approach: (i) determine the number of eligible youth students from each institution; (ii) employ the formula below to calculate representative sample size from each selected institution. (iii) a simple random sampling technique will be used to obtain the samples from each of the selected institution

$$n_i = \left(\frac{n}{N}\right)N_i$$

Where:

n = total sample size to be selected, N = total population, $N_i$ = total population of each facility, and $n_i$ = sample size from each facility.

## Data collection tools and methods

A structured questionnaire adopted from the previously studies and thereafter modified accordingly to fit the current study will be used to collect the relevant data [36, 37]. The questionnaire comprised of six parts: (i) demographic profiles, (ii) individual sexual characteristics, (iii) perception on FP use, (iv) knowledge on FP services, (v) FP services utilization, and (vi) Health provision environment for FP services. The questionnaire will be constructed in English comprising both open and closed questions.

Self-administered questionnaire will be employed to collect relevant data from the selected youth. Two pre-trained assistant data collectors will be involved in the data collection procedure. The research assistance will be the nurse practitioner. The procedure will include self-introduction by the youth, introduction of the topic of study and the purpose of the study. Then, the researchers will request their participation in the study. The questions on the data collection tool will be organized in such a way that sensitive questions will be asked later. The participants will be assured of free participation and withdrawal from the study at any time if they wish to do so. Moreover, Content and face validity will be used to assess the validity of the tool by the reviewing the literature, as well as by pilot testing the instrument prior to the study involving 10% of the actual sample size from one institution who will be excluded from the

actual study. Content validity of the instrument will be ensured by reviewing the previous study in the preparation of the questionnaire and consultation with family planning experts.

## Definition and measurement of variables

i. **Dependent variables** (Family planning utilization) The dependent variable for this study will be Family planning services utilization. Family planning utilization will be defined as the participant who will report to have ever used any type of family planning methods in the past 12 months and be able to mention them.

ii. **Independent variables** The independent variables for this study will be: socio-demographic and sexual characteristics, family planning knowledge, perception towards family planning utilization and family planning services provision environment.

## Data processing and analysis

The collected data will be cleaned and analyzed using SPSS version 25.0 software package. Descriptive analysis will be performed to present frequency distribution for demographic and sexual characteristics. Chi-square test will be performed to establish the relationship between independent variables and dependent variable. Thereafter, univariate and multivariable logistic regression will be conducted to determine the motivators for family planning service utilization among the youth. All the independent variables with p-value of less than or equal to 0.2 in binary logistic regression models will be included in multivariable logistic regression model. All the independent variables with p-value of $\leq$ to 0.05 will be regarded as statistically significant.

## Dissemination of results

The findings of this study will be disseminated to the following; University of Dodoma, authorities of the respective higher learning institution, Ministry of Health. Furthermore, the manuscript will be presented and submitted to a peer reviewed journal for publication and presented at local and international conference.

## Ethical approval and consent to participate

This study was submitted to the Directorate of Research, Publication and Consultancy of the University of Dodoma for ethical approval. The ethical committee assessed and gave the ethical approval for this study Ref. No. MA.84/261/02/'A'/91. Furthermore, permission for research conduction will be obtained from the Regional Administrative Secretary (RAS) of the Dodoma Region. Also, the researcher will present the authorization letter to introduce her to the authorities of the respective higher learning institution where the investigator is expecting to conduct the research. The individual informed consent both verbal and written will be obtained from the study participants after they have been fully informed about the study goals and the processes involved. The participants will be ensured about privacy and confidentiality. Anonymity will be maintained by the use of code number on the questionnaire instead of the participant's name and the participant will have absolute right and freedom to withdraw from the study at any time.

## Discussion

According to national surveys, including the Tanzania Demographic and Health Survey (TDHS) 2015–2016, utilization of family planning services and other SRH education remain

limited among majority of the adolescents [8]. About 68% of the women of reproductive age (WRA), including adolescents are not using FP services in Tanzania, Dodoma included [38]. Similarly, the rate of teenage pregnancy stands at 27% according to 2015/16 TDHS. By the age of 19 years, almost half (44%) of the women are either mothers or are pregnant with their first child [20]. As a result, many youth continue to engage in risky sexual practices and behaviors such as unsafe sexual partnerships and inconsistent condom use [8]. These risk behaviors damp them into teenage pregnancies which in turn results in higher risk maternal morbidity and mortality [19]. A recent scoping review reported adverse consequences of teenage pregnancy such as unsafe abortion and maternal deaths in Tanzania [39], and is considered to contribute to the national stand of 556 MMR per 100,000 live births [20], and 512 MMR per 100,000 per live births in Dodoma region, according to population and Housing Census in 2012. A study conducted in rural Tanzania found that; age, marital status, geography and health system factors; such as availability of commodities, understaffing and availability of unskilled health care providers were associated with uptake of family planning [40].

The modern contraceptive use among adolescents accounts for only 40% of the unmarried sexually active adolescents in Tanzania [41] while the study conducted in St. John University in Dodoma found that, the prevalence of family plan use was 47% among undergraduate students [21].

The consequences of not utilizing family planning services may result to adolescents not attaining their educational goals, teenage pregnancies, unwanted pregnancies that can leads to unsafe abortion and STIs including HIV/AIDs. This may also lead to morbidity and mortality and ultimately to unnecessary over use of the government expenditure for health services [4]. In Dodoma, teen pregnancy accounts for about 39% which is higher compared with other regions [42].

Therefore, the findings of this study are expected to bridge the knowledge gap by addressing motivators for family planning utilization among higher learning institution youth students in Dodoma Tanzania, which will then help to improve utilization of family planning services.

## Limitations

The following are the limitations of this study;

1. This study will be a cross-sectional study hence not useful at establishing the causal effect relationship between family planning utilization and their motivators

2. All assessments will depend on self-reports by the participants which will be likely to be a cause of informational biases.

## Supporting information

**S1 File. Ethical clearance.**
(PDF)

**S1 Questionnaire.**
(DOCX)

## Acknowledgments

Authors appreciate and thank the Ministry of Health for supporting the tuition fees and the fund for conducting this study, the University of Dodoma Research Committee for granting

ethical clearance for undertaking this study. Furthermore, the authors would like to acknowledge the School of Nursing and Public Health of the University of Dodoma for their continuous support in development of this study protocol.

## Author Contributions

**Conceptualization:** Upendo Munuo, Fabiola Vincent Moshi.

**Methodology:** Upendo Munuo, Fabiola Vincent Moshi.

**Supervision:** Fabiola Vincent Moshi.

**Writing – original draft:** Upendo Munuo.

**Writing – review & editing:** Fabiola Vincent Moshi.

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
