## [Decision Letter · Decision Letter 0]

25 Nov 2022

PONE-D-22-21995Assessing Motivators for Utilizing Family Planning Services among Youth Students in Higher Learning Institutions in Dodoma, Tanzania: Analytical Cross Sectional Study protocolPLOS ONE

Dear Dr.Upendo Munuo

Thank you for submitting your manuscript to PLOS ONE. After careful consideration, we feel that it has merit but does not fully meet PLOS ONE’s publication criteria as it currently stands. Therefore, we invite you to submit a revised version of the manuscript that addresses the points raised during the review process.

Please note the reviewers' comments and kindly attend to minor revisions required prior to publication.

We look forward to receiving your revised manuscript.

Kind regards,

Margaret Williams, Ph.D

Academic Editor

PLOS ONE

Journal Requirements:

Reviewers' comments:

Reviewer's Responses to Questions

**Comments to the Author**

1. Does the manuscript provide a valid rationale for the proposed study, with clearly identified and justified research questions?

Reviewer #1: Partly

Reviewer #2: Yes

Reviewer #3: Yes

2. Is the protocol technically sound and planned in a manner that will lead to a meaningful outcome and allow testing the stated hypotheses?

Reviewer #1: Partly

Reviewer #2: Yes

Reviewer #3: Yes

3. Is the methodology feasible and described in sufficient detail to allow the work to be replicable?

Reviewer #1: Yes

Reviewer #2: Yes

Reviewer #3: Yes

4. Have the authors described where all data underlying the findings will be made available when the study is complete?

Reviewer #1: Yes

Reviewer #2: Yes

Reviewer #3: No

5. Is the manuscript presented in an intelligible fashion and written in standard English?

Reviewer #1: No

Reviewer #2: Yes

Reviewer #3: Yes

6. Review Comments to the Author

You may also provide optional suggestions and comments to authors that they might find helpful in planning their study.

Reviewer #1: The author describes using an interviewer-administered and self-administered questionnaire. It is uncertain which method will be used. The questionnaire is not attached and its relevance cannot be assessed, neither can the modifications that the author reports to have made.

The reasons for the stratified sampling is not stated. It is also not stated how the data analysis would be adjusted to accommodate the stratification. If there are only eight higher institutions, it would have been better and simpler to include a representative sample that included all of them.

The author describes some studies on determinants of SRH and FP in Tanzania, however, the knowledge gap in the area is not clearly demonstrated. All determinants of youth SRH is indeed of importance, both motivators and disincentives. What is the existing knowledge in this regard in Tanzania or similar countries? Is there a knowledge gap? Is the problem not one of converting existing knowledge into policy and practice?

The protocol is in Standard English, however there are numerous typographical and grammatical errors.

These are underlined in the document attached

Reviewer #2: Thank you for the opportunity to review this protocol. The aim of the study is to assess motivators for utilising family planning services among youth students in higher learning institutions in Dodoma, Tanzania. The rationale behind the study is the negative of unintended and unwanted pregnancy on education goals for learners. Furthermore, youth form a significant proportion of the population in study area.

The protocol addresses and important subject area for society and institutions of higher learning. The motivation behind the study is well crafted and convincing. The objectives and aim of the study are well aligned with each other and, the methodology. The choice of the study design (cross-sectional) is reasonable. The description on how the authors arrived at the sample size, data collection and analysis tools, is clear and scientifically sound. The authors' concluding remarks are also appropriate. The protocol is well written, easy to read and to follow.

Perhaps the question one should ask is the so 'what question. If the aim was to look at barriers towards contraceptive usage among youth students in institutions of higher learning, the results would assist higher learning institutions in study and similar settings in addressing contraceptive uptake related barriers. Maybe knowledge of the motivators could have similar function- inform service providers on the packaging of contraceptive services and contraceptive related health education campaigns.

Overall, I think this is a good study with social relevance.

Reviewer #3: Recommendations:

Introduction:

Include effective family planning methods in other MLIC countries. Provide more detailes of the family planning methods that exist in Doma city.

Methods:

Include less information about the town location and more details about the four institutions where the students are studying (university, technical colleague, etc.).

I recommend to say that will be documented the characteristics of participants that won't be included in the study? To avoid bias

To report if the questionnaires have been tested for internal validity.

7. PLOS authors have the option to publish the peer review history of their article (what does this mean?). If published, this will include your full peer review and any attached files.

Reviewer #1: No

Reviewer #2: **Yes: **Lawrence Chauke

Reviewer #3: No

---

## [Author Response · Author response to Decision Letter 0]

19 Jan 2023

Reviewer 1: 

• The author describes using an interview administered and self- administered questionnaire. Which method will be used? 

RESPONSE: Self-administered questionnaire is the method that will be used during data collection.

• The questionnaire is not attached. 

RESPONSE: The questionnaire is now attached.

• The reasons for stratified sampling is not stated.

RESPONSE: Sorry, there is no stratification. Unfortunately, the word stratified in sampling was written by mistake and now it is omitted. 

• What is the existing knowledge gap in this regards in Tanzania or similar countries? 

RESPONSE: The knowledge gap is addressed

• The protocol is in standard English, however there are numerous typographical and grammatical errors. 

 RESPONSE: Typographical and grammatical errors are corrected

Reviewer 2:

Thank you for your highly valuable and appreciated inputs.

Reviewer 3:

• Provide details about the family planning methods that exist in Dodoma city.

RESPONSE: The information about the existing FP method is provided.

• Provide more information about the institutions where the students are studying.

RESPONSE: Information about the institution is provided.

• To report if the questionnaire have been tested for internal validity.

RESPONSE: The questionnaire will be tested for internal validity prior to data collection

---

## [Editor Report · Decision Letter 1]

13 Feb 2023

Assessing Motivators for Utilizing Family Planning Services among Youth Students in Higher Learning Institutions in Dodoma, Tanzania: Protocol for Analytical Cross Sectional Study

PONE-D-22-21995R1

Dear Upendo Munuo

We’re pleased to inform you that your manuscript has been judged scientifically suitable for publication and will be formally accepted for publication once it meets all outstanding technical requirements.

Kind regards,

Margaret Williams, Ph.D

Academic Editor

PLOS ONE
---

## [Editor Report · Acceptance letter]

2 Mar 2023

PONE-D-22-21995R1 

Assessing Motivators for Utilizing Family Planning Services among Youth Students in Higher Learning Institutions in Dodoma, Tanzania: Protocol for Analytical Cross Sectional Study 

Dear Dr. Munuo:

I'm pleased to inform you that your manuscript has been deemed suitable for publication in PLOS ONE. Congratulations! Your manuscript is now with our production department. 

Kind regards, 

on behalf of

Professor Margaret Williams 

Academic Editor

PLOS ONE